# Sequestration of Sulfate Anions from Groundwater by Biopolymer-Metal Composite Materials

**DOI:** 10.3390/polym12071502

**Published:** 2020-07-06

**Authors:** Md. Mehadi Hassan, Mohamed H. Mohamed, Inimfon A. Udoetok, Bernd G. K. Steiger, Lee D. Wilson

**Affiliations:** 1Department of Chemistry, University of Saskatchewan, 110 Science Place, Saskatoon, SK S7N 5C9, Canada; mehadibdbaust@gmail.com (M.M.H.); mom133@mail.usask.ca (M.H.M.); inu850@mail.usask.ca (I.A.U.); bes241@mail.usask.ca (B.G.K.S.); 2Department of Arts and Sciences, Bangladesh Army University of Science and Technology, Saidpur 5311, Bangladesh

**Keywords:** chitosan, composite materials, copper, alginate, sulfate, sorption, groundwater

## Abstract

Binary (Chitosan-Cu(II), CCu) and Ternary (Chitosan-Alginate-Cu(II), CACu) composite materials were synthesized at variable composition: CCu (1:1), CACu1 (1:1:1), CACu2 (1:2:1) and CACu3 (2:1:1). Characterization was carried out via spectroscopic (FTIR, solids C-13 NMR, XPS and Raman), thermal (differential scanning calorimetry (DSC) and TGA), XRD, point of zero charge and solvent swelling techniques. The materials’ characterization confirmed the successful preparation of the polymer-based composites, along with their variable physico-chemical and adsorption properties. Sulfate anion (sodium sulfate) adsorption from aqueous solution was demonstrated using C and CACu1 at pH 6.8 and 295 K, where the monolayer adsorption capacity (Q_m_) values were 288.1 and 371.4 mg/g, respectively, where the Sips isotherm model provided the “best-fit” for the adsorption data. Single-point sorption study on three types of groundwater samples (wells 1, 2 and 3) with variable sulfate concentration and matrix composition in the presence of composite materials reveal that CACu3 exhibited greater uptake of sulfate (Q_e_ = 81.5 mg/g; 11.5% removal) from Well-1 and CACu2 showed the lowest sulfate uptake (Q_e_ of 15.7 mg/g; 0.865% removal) from Well-3. Generally, for all groundwater samples, the binary composite material (CCu) exhibited attenuated sorption and removal efficiency relative to the ternary composite materials (CACu).

## 1. Introduction

Sulfate is one of the naturally occurring ubiquitous oxyanions commonly found in most water supplies. Sulfate contamination of water originates from both natural and anthropogenic sources. The natural sources include dissolution and oxidation of sulfate minerals such as gypsum and pyrite along with precipitation and volcanic activity. On the other hand, anthropogenic sources include sewage streams, synthetic detergents, industrial wastewater, mining drainage and mineral tailings, where levels can reach 10,000 mg/L [1,2,3,4]. The sulfate anions can exist in various forms with varying concentrations according to the water source and the nature of the industrial activities of the region [4,5]. The toxicological effects of sulfate in humans and animals have been reported to include catharsis, diarrhea, and dehydration as well as changes in methaemoglobin and sulfhaemoglobin levels [1,6]. In freshwater species, sulfate can be lethal at elevated concentrations due to osmoregulatory stress [7,8]. The latter occurs upon bacterial reduction of sulfate under anaerobic conditions to produce hydrogen sulfide (H_2_S) [9] The weak acid properties of aqueous hydrogen sulfide solution can be significantly harmful to human beings, while also raising corrosion concerns [10,11]. The presence of aqueous hydrogen sulfide is a serious toxicity concern since it does not alter the physicochemical properties (odour and taste) of water and may not be easily detected at low concentrations [11]. The consumption of a toxic dose of H_2_S in drinking water can cause dehydration, cardiac arrhythmias, amnesia, pulmonary edema, neural paralysis and other physiological problems [12,13,14]. Microbially assisted formation of H_2_S has led to water quality guidelines that recommend sulfate levels ca. 250–500 mg/L sulfate to ensure safe water for human and livestock consumption [6,15,16].

The toxicity challenges presented by the presence of sulfate species in water along with its high solubility and mobility calls for an efficient and cost-effective strategy for its controlled removal from groundwater as well as industrial effluent [17,18]. Techniques for sulfate removal such as precipitation [1,19], biological degradation [20,21], reverse osmosis [22], membrane filtration [23], electrocoagulation [24], crystallization [25], adsorption [26] and/or ion exchange [27], are commonly known methods. Among these techniques, adsorption-based removal with biopolymer adsorbents may offer advantages (minimal infrastructure and operational cost) over synthetic adsorbent materials or conventional treatment methods such as reverse osmosis [26]. As well, the synthetic utility of biopolymers for tailored modification add to its technical feasibility, cost effectiveness, sustainability, selectivity and treatment efficiency for diverse effluent sources with variable sulfate concentration [28,29]. Additionally, the aforementioned properties of biopolymers lend to their suitability for various environmental applications that include but are not limited to flocculation, drug delivery and agriculture [30,31,32,33,34,35]. By contrast, conventional materials such as zeolites are naturally occurring minerals, however; the production and regeneration costs may pose limitations on their sustainable utility as adsorbents. By contrast, renewable biomass such as alginate and chitosan continue to generate great interest due to their relative abundance and synthetic versatility [36,37,38].

Chitosan has found many applications that build upon its unique properties such as biocompatibility, lack of odour and biodegradability [39]. In particular, chitosan is a partially deacetylated form of chitin that is characterized by available amino and hydroxyl groups. At pH conditions below 6.5, the amino groups of chitosan are protonated, which results in variable water solubility according to the type of acid media and the degree of deacetylation [40]. The –OH and –NH_2_ groups of chitosan offer potential ligation sites to form complexes with transition metal ions [41,42,43] along with anionic biopolymers such as alginate [44,45]. On the other hand, alginate is an anionic biopolymer obtained from brown algae and bacteria where, α-L-guluronic (G) and β-D-mannuronic (M) acid constitute its monomer units [44,46]. The formation of chitosan- or alginate-metal ion binary complexes afford modification of the physicochemical and structural properties of the biopolymer or metal ion precursors upon complex formation due to changes in electrostatic potential. The unique properties of such complexes has led to their utility as dialysis and ultrafiltration membranes [47], drug delivery agents [48] and biosorbent materials [49]. Recent review articles [39,50,51,52] on alginate and chitosan-based adsorbents reveal their application as adsorbents for cation and anion species. As well, such biopolymers are established for removal of dyes and heavy metals, whereas studies of biopolymer-metal ion complexes are sparsely reported for the adsorption of sulfate from aqueous solution, according to the examples provided in Table 1.

This study focuses on the synthesis and characterization of binary (two-component) and ternary (three-component) composite materials that contain animal and/or plant biopolymers (chitosan and alginate) as building blocks in conjunction with copper (II) species. The unique contribution of Cu(II) species to the formation the ternary biopolymer composites yield enhanced adsorption capacity of sulfate anions for laboratory and groundwater samples will be demonstrated herein. A comparison of related materials in this study will reveal that the composition of composites (binary and ternary) and their sulfate uptake properties are unique at pH 7.

## 2. Materials and Methods

### 2.1. Materials

Low molecular weight industrial grade chitosan (C) powder with ~75–85% deacetylation, copper (II) chloride (≥97%), barium chloride (≥99.9%), KBr (spectroscopic grade), sodium hydroxide, HCl, sodium chloride (≥99.9% crystalline), sodium sulfate anhydrous (≥99.9%) and acetic acid were obtained from Sigma Aldrich (St. Louis, MO, USA). Alginic (A) acid sodium salt (low viscosity) was obtained from Alfa Aesar (Tewksbury, MA, USA). All materials were used as received without further purification.

#### 2.1.1. Synthesis of a Chitosan Copper (II) Binary Composite: CCu

Chitosan (1 g) was dissolved in 50 mL (5% acetic acid) solution. The pH of the solution was raised to pH 5.6 using 1M NaOH. CuCl_2_ solution (0.006 mol) was then added dropwise to the chitosan solution and incubated for 3 h, where the final solution pH was near pH 7. The pH of the resulting mixture was adjusted to 7 with 2M NaOH to precipitate the chitosan-copper (CCu) binary composite. The composite was washed with copious amount of Millipore water to remove the unreacted reagents. The product was dried in an oven at 60 °C and the composite was ground in a mortar and pestle and passed through a 40-mesh sieve prior to use.

#### 2.1.2. Synthesis of Chitosan, Alginate and Copper-Based Ternary Composites: CACu

The synthesis of CACu1 ternary composite was carried out according to the procedure reported in Section 2.2.1 with minor modification, as follows: 1 g (0.006 mol) of chitosan was dissolved in 50 mL of 5% acetic acid solution, while 1.3 g (0.006 mol) of alginate (A) was dissolved in 50 mL of Millipore water. The pH of the solution was adjusted to pH 5.6 using 1M NaOH added dropwise to the alginate solution with stirring until a homogeneous blend was obtained. CuCl_2_ solution (0.006 mol) was added to the chitosan-alginate blend with stirring and incubated for 3 h. The pH of the mixture was adjusted to 7 with 2M NaOH to precipitate the chitosan-alginate-copper (CACu1) ternary composite. The composite was washed with a generous amount of Millipore water to get rid of unreacted reagents. The product was dried in an oven at 60 °C and the composite was ground in a mortar and pestle and passed through a 40-mesh sieve prior to use. The other composite materials CACu2 and CACu3 were synthesized according to the above procedure, where the weight of chitosan and alginate were varied to yield the desired weight ratio (cf. Table 2).

### 2.2. Characterization

#### 2.2.1. Fourier Transform Infrared (FTIR) Spectroscopy

The FTIR spectra of the chitosan (C), alginate (A), CCu and CACu composite materials were obtained using a Bio-RAD FTS-40 IR spectrophotometer (Bio-Rad Laboratories, Inc., Philadelphia, PA, USA). Dried powder samples were mixed with pure spectroscopic grade KBr in a 1:10 weight ratio with grinding in a small mortar and pestle. The DRIFT (Diffuse Reflectance Infrared Fourier Transform) spectra were obtained in reflectance mode at 295 K with a resolution of 4 cm^−1^ over a spectral range of 400–4000 cm^−1^. Multiple scans were recorded and corrected relative to a KBr background.

#### 2.2.2. X-ray diffraction (XRD)

X-ray diffraction (XRD) profiles of the composite materials were obtained using a Rigaku Ultima IV X-Ray diffractometer, Cu-source, wavelength 1.54056Å (Austin, TX, USA) in the range of 2θ = 10°–80° at 40 kV potential and 40 mA current.

#### 2.2.3. Thermal Gravimetric Analysis (TGA)

Thermal profiles of the composite materials were obtained using a TA Instruments Q50 TGA system (New Castle, DE, USA) using a heating rate of 5 °C/min up to 500 °C and nitrogen as the carrier gas. The results reported herein are shown as first derivative (DTG) plots of weight with temperature (%/°C) against temperature (°C).

#### 2.2.4. Solids C-13 NMR Spectroscopy

A Bruker AVANCE III HD spectrometer (Bruker Bio Spin Corp., Billerica, MA, USA) furnished with a 4 mm DOTY CP-MAS (cross-polarization with magic angle spinning) solids probe operating at 125.8 MHz (1H frequency at 500.2 MHz) was used to acquire the C-13 solids NMR spectra of the chitosan, alginate and the composite materials. The experimental conditions are given as follows: spinning speed of 10 kHz, a ^1^H 90° pulse of 3.5 µs, a contact time of 0.75 ms, with a ramp pulse on the ^1^H channel, MAS rate of 10 kHz, a ^13^C 90° pulse width of 3.15 µs and a 25 kHz SPINAL-64 ^1^H decoupling sequence during acquisition. In general, 600–5000 scans were accumulated, with a recycle delay of 2 s. All experiments were recorded using a 71 kHz SPINAL-64 decoupling sequence during acquisition, and the ^13^C NMR chemical shifts were referenced to adamantane at 38.48 ppm (low field signal).

#### 2.2.5. Raman Spectroscopy

One-dimensional (1–D) Raman spectra were obtained using a Renishaw InVia Reflex Raman microscope (785 nm solid state diode laser with a 1200 lines/mm grating system) (Renishaw plc, New Mills, UK) with a Pelletier cooled CCD (charge coupled device) detector (400 × 576 pixels). The instrument wavelength was calibrated at 520 cm^−1^ using an internal Si (110) sample.

#### 2.2.6. Differential Scanning Calorimetry (DSC)

DSC thermograms were acquired using a TA Q50 (New Castle, DE, USA) thermal analyzer over a variable temperature range (30–180 °C). The scan rate was set to 10 °C/min while nitrogen gas was used to regulate the sample temperature and purge the sample compartment housing, where ca. 10 mg of sample was used for analysis.

#### 2.2.7. Point-of-Zero-Charge (PZC)

The point-of-zero-charge (PZC) of sample materials was determined according to a reported method [59]. A stock solution of NaCl (0.01 M) was prepared and 25 mL portions were transferred into five vials (8-dram). The solution pH conditions of the samples were adjusted between pH 2 to 10 using aqueous NaOH or HCl solutions. The sorbent materials (ca. 50 mg) were added to each solution and allowed to equilibrate for 48 h before the final pH was recorded.

#### 2.2.8. Solvent Swelling Test

The solvent swelling properties in water of the composite materials (ca. 100 mg) was measured upon equilibration in Millipore water (30 mL) for 24 h. The degree of swelling (*S_w_*) in water for the composites was calculated according to Equation (1):(1)$$S_{w}=\frac{W_{s}-W_{d}}{W_{d}}\times 100\%$$

*W_s_* is the wet sample weight, and *W_d_* refers to the dry sample weight after oven drying at 60 °C for 24 h to a constant value.

#### 2.2.9. X-ray Photoelectron Spectroscopy (XPS)

All XPS measurements were collected using a Kratos (Manchester, UK) AXIS Supra system equipped with a 500 mm Rowland circle monochromated Al K-α (1486.6 eV) source, combined hemi-spherical analyzer (HSA) and spherical mirror analyzer (SMA). A spot size of hybrid slot (300 × 700 microns) was used. All survey scan spectra were collected in the 5 to 1200 binding energy range in 1 eV steps with a pass energy of 160 eV. An accelerating voltage of 15 keV and an emission current of 15 mA were used for the XPS data collection.

#### 2.2.10. Leaching of Cu(II) Species

For the leaching test, the adsorbent (ca. 100 mg) was added to 30 mL Millipore water and equilibrated at 295 K on a horizontal shaker table for 24 h to determine whether the Cu(II) concentration exceeded 1000 µg/L, according to measurements acquired using ICP-OES.

### 2.3. Adsorption Studies

#### 2.3.1. Equilibrium Adsorption Experiments

The equilibrium adsorption studies were carried out in batch mode using Na_2_SO_4_ (aq) at variable concentration (100–5000 ppm) as the adsorbate system. A fixed amount of adsorbent (~10 mg) was added to 6-dram vials and 15 mL of Na_2_SO_4_ (aq) solution at variable concentration at pH 6.8 was transferred to the vials. The mixtures were equilibrated at 295 K on a horizontal shaker table for 24 h. The initial concentration (C_0_) before sorption and residual concentration after sorption were determined using a Thermo Scientific^TM^ SPECTRONIC 200E (Ottawa, ON, Canada) spectrophotometer at 420 nm. The samples were centrifuged prior to UV-vis spectroscopy analyses as required. The uptake of sulfate ions by the adsorbents was determined according to Equation (2). For the ground water samples, ~100 mg of the composite materials and 30 mL of the groundwater samples were used for the batch mode experiments.
(2)$$Q_{e}=\frac{C_{0}-C_{e}}{m}\;V$$

Q_e_ is the quantity of adsorbed species (SO_4_^2−^) in the solid phase at equilibrium (mg/g); C_0_ is initial concentration of the adsorbate (mg/L) in solution; C_e_ is concentration of SO_4_^2−^ at equilibrium (mg/L) in aqueous solution; V is volume of adsorbate solution (L) and m is the weight (g) of sorbent. Herein, four adsorbents were used for the removal of sulfate ions for a synthetically prepared solution, along with groundwater samples labeled as Well-1, -2 and -3. The elemental analyses of these water samples are listed in Appendix A, respectively, as listed in the Appendix A.

#### 2.3.2. Sorption Isotherms and Modeling

The adsorption results were analyzed using the Sips isotherm model (Equation (3)) where the “best fit” of Equation (3) to the data was obtained by minimizing the SSE (Equation (4)) for all data across the range of conditions [60]. Q_ei_ is the experimental value, Q_ef_ is the calculated value from the data fitting and N is the number of Q_e_ data points.
(3)$$Q_{e}=\frac{Q_{m}{\left(K_{s}C_{e}\right)}^{n_{s}}}{1+{\left(K_{s}C_{e}\right)}^{n_{s}}}$$
(4)$$\mathrm{SSE}=\sqrt{\frac{{\left(Q_{\mathrm{ei}}-Q_{\mathrm{ef}}\right)}^{2}}{N}}$$

## 3. Results and Discussion

The structural characterization of the composites is described in the following sections by several complementary spectroscopic, thermal analysis and other methods. As well, the sulfate adsorption properties were studied using laboratory and groundwater field samples at variable conditions.

### 3.1. Fourier Transform Infrared (FTIR) Spectral Analysis

Functional groups and chemical bonds of a molecule can be characterized by use of IR spectral results. Figure 1 shows the DRIFT spectra of the biopolymers and composites (C, A, CACu1 CACu2, CACu3 and CCu) over the 4000 to 500 cm^−1^ spectral region. The IR spectrum of pristine chitosan has a broad band at 3473 cm^−1^, which is assigned to the N–H and hydrogen bonded O–H vibrational bands. The absence of a sharp absorption band near 3500 cm^−1^ in all composite materials indicates the relative absence of free OH groups. Moreover, the C–H stretch region had a high intensity (2898 cm^−1^) and lower intensity band (2877 cm^−1^) assigned to the symmetric and asymmetric modes of the CH_2_ groups, respectively [61]. A characteristic CH_2_ scissoring band is present at 1418 cm^−1^ along with a C=O stretch of an amide band at 1663 cm^−1^ for chitosan, where the bending vibrations of a secondary amide (δ_N–H_) is noted at 1596 and 1541 cm^−1^ [62]. The IR spectrum of sodium alginate has a characteristic –OH stretching band at 3480 cm^−1^, C–H stretching vibrations at 2927 cm^−1^ and asymmetric stretching vibration for –COOH at 1648 cm^−1^. The bands at 1306 and 1426 cm^−1^ relate to symmetric C–O stretching of the carboxyl groups, whereas the IR bands at 937 and 1104 cm^−1^ were assigned to elongation of C–O groups [63,64,65]. Upon composite formation, the C=O stretching and bending bands exhibit broadening along with a shift to lower wavenumber values, respectively. The broadening and shifting of the IR bands indicate the involvement of the C=O groups in composite formation between chitosan and alginate along with Cu(II) species [66]. The IR results indicate the coordination of amine groups and the amide C=O groups with Cu(II) species yield the formation of composites. However, in the case of CACu2, asymmetric and symmetric stretching vibrations for the –COOH groups of alginate did not reveal any significant change. Thus, these results suggest that limited coordination occurs between alginate and the Cu(II) species upon ternary composite formation.

### 3.2. Raman Spectroscopy

While IR spectroscopy provides insight on functional groups that undergo a change in the dipole moment, Raman spectroscopy is complementary due to its sensitivity to transitions that undergo a change in polarizability. The Raman spectral data of the biopolymers and their composites is inferred to provide additional information on the binary and ternary composites reported herein. For example, in the case of IR spectra, it is evident that the amide (C=O) and amine groups undergo coordination with Cu(II) species to form both binary and ternary composites. The Raman spectra of chitosan, alginate and the films for the composite materials CCu, CACu1, CACu2 and CACu3 are shown in Figure 2. The characteristic Raman bands that reveal an interaction between Cu(II) species ion and carbonyl groups (C=O) of chitosan for the composites are observed between 281 to 349 cm^−1^ [67,68,69]. The formation of composites is also supported by the DRIFTS results, as evidenced by broadening of the C=O stretching band of amide groups between 1647 and 1655 cm^−1^ (cf. Figure 1). In the case of the CACu1, characteristic Raman spectral shifts are noted for interactions between the Cu(II) species and the carbonyl groups of chitosan but were not clearly observed for the other composites as noted in Figure 2. Additionally, Raman bands at 219/226/230/422/439/441/442/443 cm^−1^ may relate to the interaction between Cu(I) species and the amide carbonyl group (C=O) of chitosan for the composite materials [70].

The Raman results provide support that reduction of Cu(II) species to Cu(I) species may occur upon binding with the C=O groups of chitosan. A noteworthy observation is the absence of obvious signatures related to interactions between the carboxyl groups of alginate and Cu(II) species in the Raman spectra. The absence of such bands provide support that coordination occurs between Cu(II) species with chitosan. This may be due to steric interactions that occur due to hydration of the carboxylate anion that screens interactions with the Cu(II) species. In the case of the C=O groups of chitosan, a more apolar environment exists at ambient pH conditions that may favour complexation of chitosan with Cu(II) species versus the –COO– groups of alginate.

### 3.3. Thermogravimetric Analysis (TGA)

Thermal stability, resistance to oxidation and physical phenomena that show weight-loss events relates to the compositional nature of copolymer materials which can be studied from TGA-Differential Thermal Analysis (TGA-DTA) profiles [71,72]. Moreover, TGA is particularly sensitive for the detection of differences in structurally similar materials and complex multicomponent systems that vary according to their composition. The latter case applies to the composite materials reported herein. The thermal decomposition behavior of chitosan, alginate, CCu1 and the ternary composite materials are illustrated in Figure 3 and Appendix A. These profiles reveal two or three thermal events that vary across a temperature range up to 400 °C, in parallel agreement with a related study [73]. In general, the first thermal event occurs between 36 to 120 °C that relates to loss of bound surface water and/or entrapped water from the composite network [74]. The first thermal event (36 to 120 °C) for the composite materials reveal a large fractional weight loss (8.2%) for CACu2, whereas 3.0% was observed for CACu2. This affirms the presence of abundant polar groups in CAlCu3 that affirm the formation of hydrogen bonds with water relative to CACu2. However, the weight loss of CCu and CACu1 was similar (6.9% and 7.2% respectively), as noted in Appendix A. The thermal events above 150 °C relate to decomposition of the polymer framework. In the case of the biopolymers (C and A), thermal event occur at different *T*_max_ values: 295 °C for C and 247 °C for A. These thermal events relate to the depolymerization and dehydration of the saccharide rings [75]. In the case of CCu, the weight loss events at ~215 °C result due to the breakdown of the biopolymer framework, whereas the thermal event at ~267 °C relates to bond breaking between Cu(II) species and the carbonyl group (C=O) of chitosan. The thermal event between 170–350 °C for CACu1, ca.180–320 °C for CACu2 and CACu3 materials relate to decomposition of C and A domains, along with bond breaking between Cu(II) ions and the carbonyl (C=O) group of chitosan, respectively. The lowest onset degradation temperature displayed by CACu2 may arise due to the higher alginate content in the composite; thereby resulting in a weaker interaction of Cu(II) species with the carbonyl group of chitosan. By comparison, CACu2 and CACu1 composites exhibit lower thermal stability relative to CCu and CACu3, in agreement with the more efficient binding of Cu(II) species to chitosan.

### 3.4. Solid State ^13^C NMR Spectroscopy

Solid state ^13^C NMR spectroscopy allows for the elucidation of changes in the chemical environment of biopolymer materials due to changes in the chemical shifts of nuclei that undergosolvation or interactions with other chemical species. The spectral assignment of chitosan in Figure 4 is given: δ_105.1_ = C-1; δ_57.1_ = C-2; δ_74.9_ = C-3/C-5; δ_83.1_ = C-4 and δ_61.1_ = C-6. Since the deacetylation of chitosan (~75–85%) is moderate, the acetyl group signature is noted at δ_173.9_ = C_7_ and δ_23.6_ = C_8_ [76]. The spectral assignment for alginate is also provided: δ_101.1_ for C-1; the four ring carbon (C-2, C-3, C-4 and C-5) lines at δ_59.87_ to δ_85.3_, where the characteristic C=O group appears at δ_175.4_. In the spectra of CACu1, CACu2, CACu3 and CCu composite materials, the merging of the signals for both carbonyl groups (C-7 for chitosan and C-6 for alginate) along with a broader ^13^C NMR signature at δ_59.87–85.3_ are observed. A new upfield ^13^C NMR line occurs for all composites near the δ_28.7-35.9_ region. These signals appear at δ_31.4,_ δ_30.1,_ δ_30.8_ and δ_31.7_ for CACu1, CACu2, CACu3 and CCu, respectively. These new peaks provide evidence of effective interaction between the copper species and the carbonyl oxygen of chitosan rather than interaction between Cu(II) with the carboxylate group of alginate, in agreement with the Raman spectral results in Figure 2. Moreover, the C-8 peak of chitosan becomes broadened for all composites, in agreement with the interaction of Cu(II) species and the C=O (acetyl) group of chitosan. In addition, ring carbon signatures for chitosan and alginate show broadening in the composites upon complex formation between the precursor units (Cu(II), chitosan and alginate). The ^13^C NMR spectra (Figure 4) and other results (cf. Figure 1, Figure 2 and Figure 3) provide support that interactions occur between Cu(II), chitosan and alginate, where the Cu(II) species are bound at the C=O (acetyl) moiety of chitosan, in agreement with the TGA and Raman results.

### 3.5. Differential Scanning Calorimetry (DSC)

DSC studies were carried out to identify the effect of structural differences on the hydration of C, CCu and CACu composite materials since hydration effects are known to play a key role in the adsorption properties of biopolymer materials. The DSC results are summarized in Table 3 along with the accompanying thermograms in Figure 5.

In general, all composite materials display one endothermic peak within the range of 30–180 °C with *T*_max_ values between 97.5–146.6 °C while onset temperatures resided between 51.9–120.9 °C. The endotherms reveal characteristic features of water desorption that concur with the temperature range observed by TGA analysis. Among the four composites, CACu2 showed the lowest *T*_max_ value of 125.3 °C while CACu3 displays the highest the *T*_max_ value (146.6 °C). The maximum Δ*H_des_* value (2016 J g^−1^) for CACu1 composite material provides evidence of strongly bound water with the active sites of the biopolymers and/or with Cu(II) species. Based on these results, a higher portion of bound water can be found in the CACu1 composite versus the other composites. The DSC results provide support that accounts for trends in water swelling at equilibrium that are further described in Section 3.6.

### 3.6. Swelling Tests

The relative hydrophile-lipophile balance (HLB) of chitosan and the composite materials can be estimated via solvent swelling (*S_w_*) in water at equilibrium conditions. The estimated *S_w_* values for the composites were reported as a triplicate average at ambient pH and 23 °C. The values of *S_w_* provide an account of the accessibility of polar functional groups on the biopolymer surface. It can be inferred that the level of crosslinking and/or interaction between Cu(II) species and chitosan likely play an important role in the overall hydrophilicity of the composite materials. CCu showed the lowest swelling ratio (*S_w_* = 87.4%) among the composites due to more effective Cu(II) coordination with chitosan. Similarly, greater crosslinking between the biopolymers (chitosan/alginate) is also likely to influence the relative accessibility of the polar functional groups of the composites. In turn, reduced accessibility of polar groups contribute to a lowering of the *S_w_* values. An increased amount of alginate (beyond the stoichiometric equivalency) in the case of CACu2 led to a slightly higher *S_w_* (91.37%) value relative to CCu, in agreement with the DSC results. The role of incomplete crosslinking or the accessibility of active functional groups for Cu(II) species in CACu2 material contributes to its slightly higher *S_w_* ratio relative to CCu. By comparison, CACu1 and CACu3 showed higher *S_w_* values (148.6% and 162.6%, respectively) which concur with an observed stronger interaction of Cu(II) species with the active sites of chitosan that account for greater hydration and increased *S_w_* values for these composite materials.

### 3.7. X-ray Photoelectron Spectroscopy (XPS)

Atomic composition (At. %) of CACu1 was studied using XPS after exposure to the various groundwater samples (Well-1, Well-2 and Well-3) that contain SO_4_^2−^ ions. The XPS results are shown in Appendix A and Table 4. In general, comparable C, N, O and Cu bands were observed with acceptable atom content (At. %) in Table 4 that indicate negligible leaching of Cu(II) ions occur from the composites into water. The wide scan spectra in Appendix A indicate that the composition of the sorbent does not change markedly before and after the adsorption process. However, no pronounced sulfur signature was observed in the XPS spectra for CACu1.

### 3.8. X-ray Diffraction (XRD) Results

X-ray diffraction (XRD) is a complementary structural method that can provide insight on the atomic arrangement or the crystalline nature of biopolymers. X-ray diffraction provides insight on the lattice arrangement and is dependent on the scattering of X-rays by the electron density of atoms, and the crystalline nature of multi-component materials. XRD studies of powdered chitosan show two broad signatures at 2θ = 13° and 27° (cf. Appendix A). These XRD features provide support that chitosan exists in a semi-crystalline form [77]. By contrast, alginate shows a comparatively sharp band at 2θ = 32°. However, no significant broad and sharp XRD bands are observed from the profiles of CCu- and the CACu-based composites. These results suggest that the composite materials possess more amorphous structural features relative to that of chitosan in its semi-crystalline state. This trend compares well with the anticipated amorphous composite structure for such multi-component biopolymer systems.

### 3.9. Sorption Studies

Adsorption isotherm studies enable an investigation on the role of functional groups, surface structure, morphology and chemical composition of binary and ternary composite materials that contain Cu(II) species. The variation in composite structure in relation to the uptake of sulfate is inferred to provide insight on the structure-adsorption properties of binary and ternary systems. In this work, the sorption capacities of the composite materials were studied with a synthetic sodium sulfate solution to assess their inherent sulfate removal properties. In addition, three Saskatchewan groundwater samples with a multitude of potentially interfering anions was investigated to ascertain the sulfate adsorption properties in environmental matrices of variable composition. The equilibrium adsorption isotherms of sulfate species with powdered chitosan and CACu1 are shown in Figure 6. For both sorbents, the adsorption of sulfate increases nonlinearly as C_e_ increases, until the surface-active sites become saturated at elevated sulfate levels, as evidenced by the appearance of the plateau region in the isotherm.

Unmodified chitosan powder has comparatively lower adsorption than the CACu1 composite. The trend can be accounted for by the increase in surface area (SA) and availability of favourable adsorption sites on the CACu1 composite material. The trend may relate to the presence of Cu(II) along with the greater amorphous nature of such ternary composites versus the semi-crystalline properties of chitosan (cf. XRD pattern in Appendix A). The isotherm parameters are well described by the Sips model according to the best-fit results (R^2^ = 0.98 to 0.99 for C and CACu1), where the best-fit parameters are summarized in Appendix A. At pH 7, a comparison of the adsorption capacity Q_e_ (mg/g) of various sorbents for SO_4_^2−^ anion removal from the aqueous phase are listed in Table 3. According to the data in Table 5, CACu1 showed greater adsorption capacity (Q_m_ = 371.4 mg/g) at pH 6.8 and 295 K relative to other reported adsorbents adsorbents (cf. Table 6). Furthermore, to assess the suitability of the composite materials for practical removal of sulfate anions from water, the sorption capacity of C and CACu1, along with three chitosan-based materials (CACu2, CACu3 and CCu) were independently evaluated with three groundwater (Well-1, -2 and -3) samples. The variable concentration of sulfate ions and other chemical components for the groundwater sample are listed in Appendix A.

Figure 7a–f display the results for Q_e_ (mg/g) and removal efficiency (%) of sulfate ions for the various composite materials. A comparison of the results reveal that the uptake (Q_e_; mg/g) of sulfate for the CACu1 composite was greatest (Q_e_ = 77.1 mg/g) for the Well-3 sample. This result suggests that the presence of Cu(II) species in the chitosan-alginate network enhanced the affinity toward SO_4_^2−^ ions. Among the ternary composites, CACu3 displayed the best sorption capacity for Well-1 and Well-2 samples, whereas unmodified chitosan displays the lowest SO_4_^2−^ removal of for the Well-3 sample. This may be explained by the composition of the different well water samples with respect to effects due to the role of competing anions in the groundwater samples, as shown in Table 5.

The comparison of groundwater in Table 5 reveals variation in sulfate concentration, overall hardness and matrix anion composition of the samples. The variance in anion compositions provides an account of competitive binding to the adsorption sites of the biopolymer materials. CACu1 showed the lowest sulfate uptake capacity for Well-2, where the nitrate concentration is notably higher relative to Well-1 and -3 samples. Greater uptake of sulfate by CACu1 for the Well-3 sample parallels its greater sulfate content. CACu2 shows low uptake for Well-2 and -3 samples irrespective of the sulfate concentration, indicating that other anions exhibit greater affinity with this composite material. CACu3 exhibited the highest capacity at the lowest sulfate concentration in conjunction with the lowest chloride and bicarbonate concentration in the Well-1 sample. The lowest performance was observed in Well-3 sample despite its 3-fold-greater sulfate concentration as in the Well-1 sample. This trend may indicate that bicarbonate and nitrate are competitor ions for the active sites, despite the increase in sulfate concentration. Also, CCu showed the highest sulfate uptake capacity with the lowest concentration of nitrate, bicarbonate and chloride, whereas the Well-3 sample had marginally higher uptake than the Well-2 sample. This trend indicates that bicarbonate, and nitrate show competitive ion adsorption with sulfate. Regarding the influence of Cu(II) chloride or sulfate salts, it has been reported that sulfate anions tend to preferentially exchange with chloride [78]. The sulfate uptake properties by the various biopolymer materials differ significantly with regards to the matrix composition of the water samples, and this should be considered for practical applications of such composite materials for sulfate uptake in different aquatic environments.

Based on the characterization results reported herein, an adsorption mechanism for the removal of sulfate ions from the groundwater is proposed to involve electrostatic attractions between the sulfate ions and the free amine groups of chitosan as well as the accessible sites of complexed Cu(II) species within the composite material (cf. Figure 8). While leaching of Cu(II) can pose a risk to aquatic organisms, the ICP-OES results (cf. Appendix A) show that minor Cu(II) leaching from the biopolymer materials was observed. It is noted that the presence of Cu(II) in the aqueous phase remained below 1000 µg/L. This value is below the drinking water regulatory standards set forth by the Saskatchewan Health Authority (cf. Appendix A) and provides additional support for the use of Cu(II) in the biopolymer composites reported herein for practical applications.

## 4. Conclusions

Binary chitosan-Cu(II) (CCu)- and ternary chitosan-alginate-Cu(II) (CACu)-based composite materials were synthesized at variable composition. The products were characterized by several complementary methods: spectroscopic (FTIR, solids C-13 NMR, XPS and Raman), thermal analysis (DSC and TGA), XRD, point-of-zero-charge and solvent swelling. The incorporation of Cu(II) species into the chitosan-alginate composite was supported by the materials’ characterization results. Batch adsorption reveals a favourable adsorption of sulfate anions in laboratory-prepared samples of sodium sulfate, along with adsorption of sulfate anions in a complex groundwater matrix. The adsorption data are well described by the Sips isotherm model, where CACu1 had a Q_m_ value of 371.4 mg/g, notably higher than a selection of reported literature values (cf. Table 6). Under similar reaction conditions, various ternary chitosan-alginate-Cu(II)-containing adsorbents displayed variable sorption capacities for sulfate anions in groundwater. CACu3 displays the best uptake properties (Q_e_ = 81.5 mg/g; 11.9% removal efficiency) among the adsorbents reported herein. The results obtained in this study provide support on the role of Cu(II) species in stabilizing the chitosan-alginate network that additionally favour sulfate adsorption for environmental water samples. This research highlights a green strategy for tuning the adsorption properties of binary and ternary composite materials that contain chitosan and alginate along with Cu(II) species to afford enhanced uptake of sulfate ions from laboratory and groundwater samples. Such types of binary and ternary biopolymer-based adsorbents have practical utility as green adsorbents for the treatment of industrial wastewater and other types of anion pollutants in aquatic environments. The unique contribution of Cu(II) species in the biopolymer framework of composite materials is shown via a modular synthetic strategy that yield enhanced adsorption properties (cf. Table 5) toward sulfate anions.

## Figures and Tables

**Figure 1 polymers-12-01502-f001:**
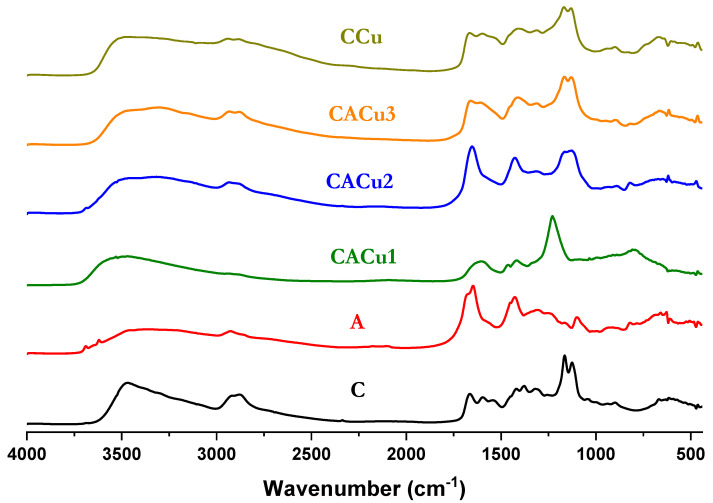
FTIR spectra of chitosan (C) and alginate (A) biopolymers, and its composite forms: CACu1, CACu2, CACu3 (ternary) and CCu (binary) materials.

**Figure 2 polymers-12-01502-f002:**
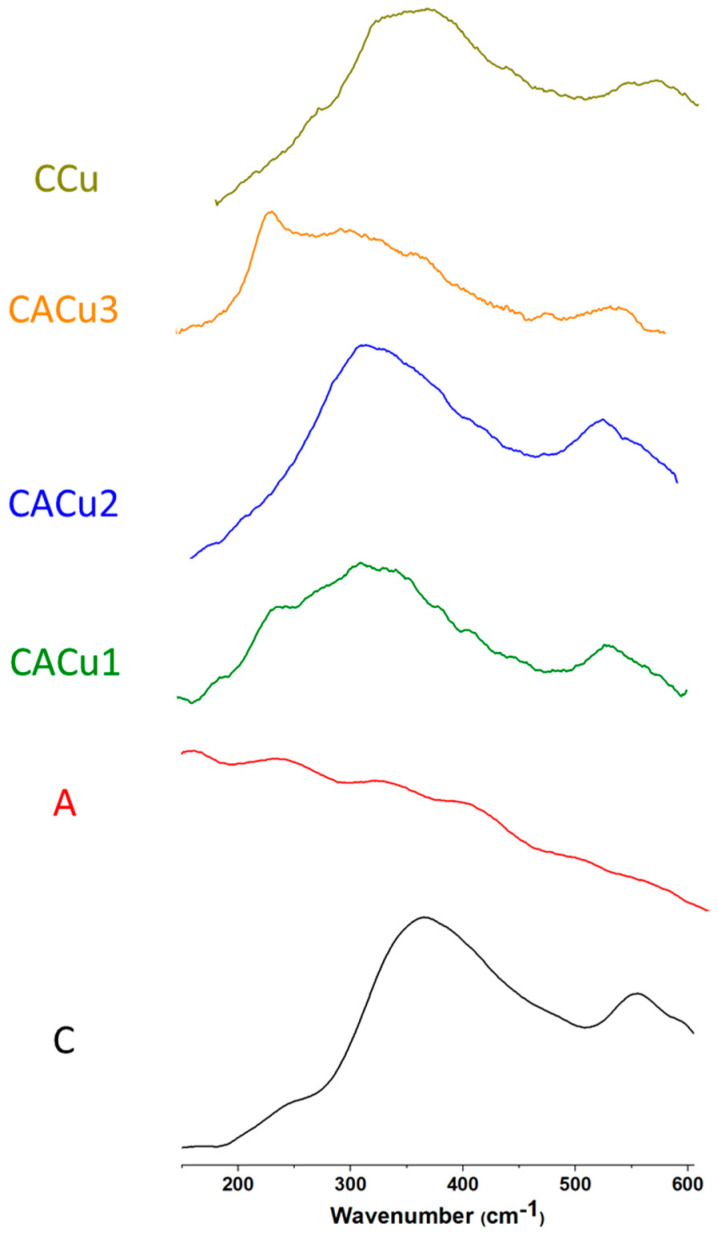
Raman spectra of the biopolymers (C, A), ternary composites (CACu1, CACu2, CACu3) and binary composite (CCu) materials.

**Figure 3 polymers-12-01502-f003:**
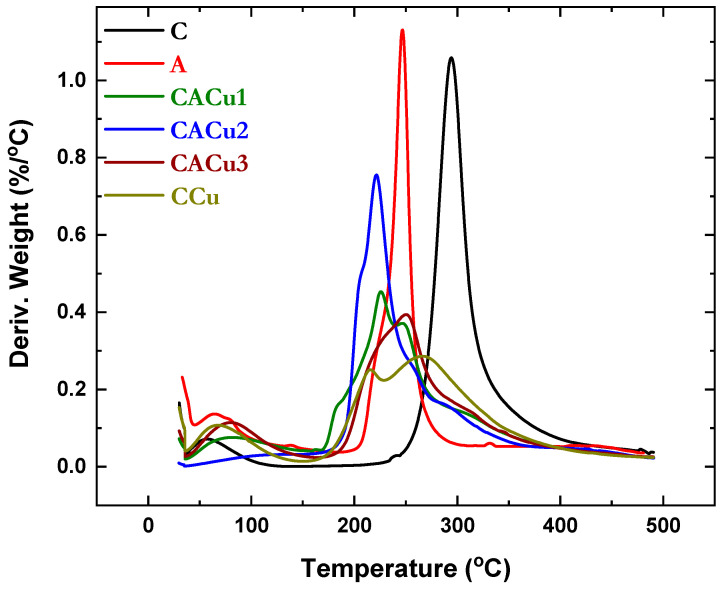
Derivative plot from TGA for the biopolymers (C, A), ternary composites (CACu1, CACu2, CACu3) and a binary composite (CCu) material.

**Figure 4 polymers-12-01502-f004:**
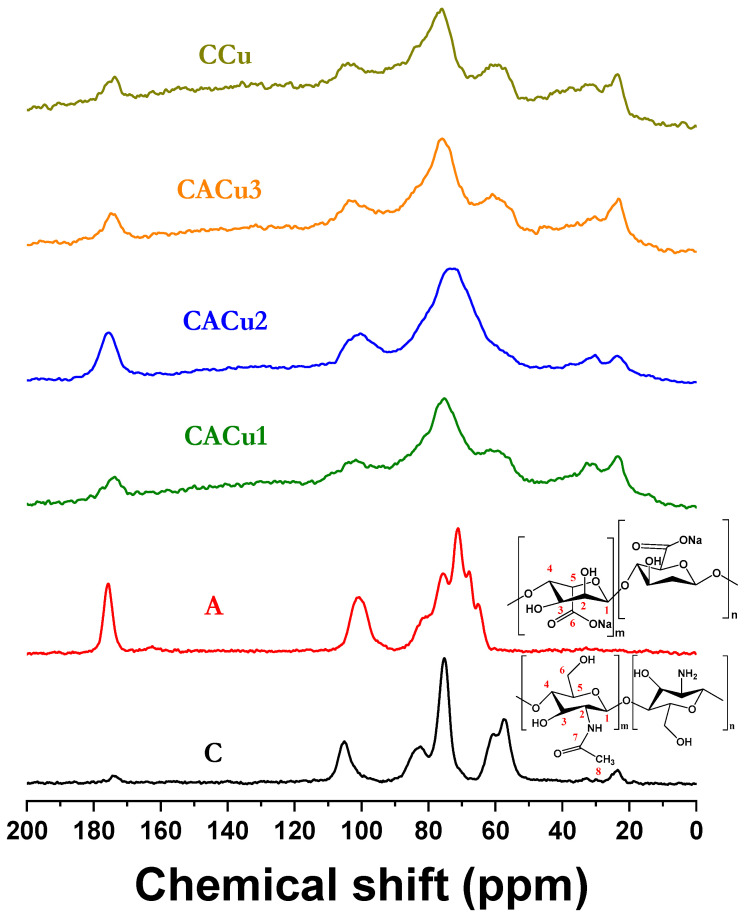
Solid state C-13 NMR of the biopolymers (C, A), binary (CCu) and ternary (CACu1, CACu2, CACu3) composites.

**Figure 5 polymers-12-01502-f005:**
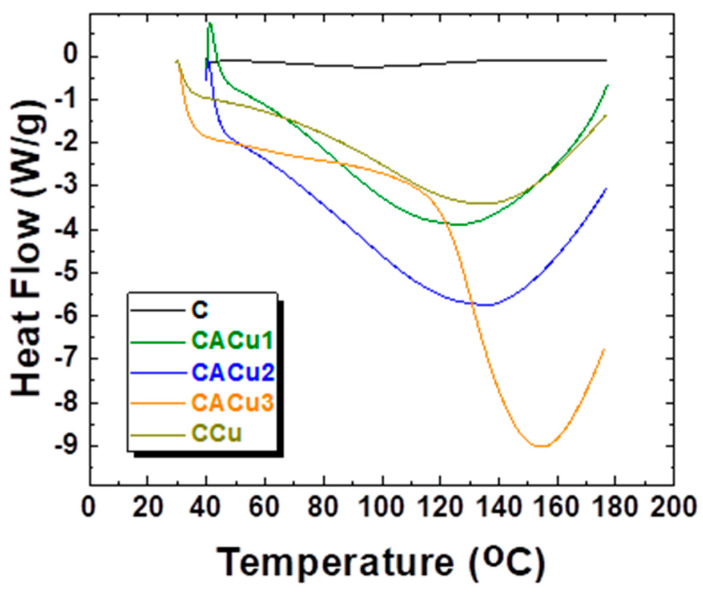
DSC of chitosan (C)), ternary composites (CACu1, CACu2, CACu3) and binary composite (CCu) materials.

**Figure 6 polymers-12-01502-f006:**
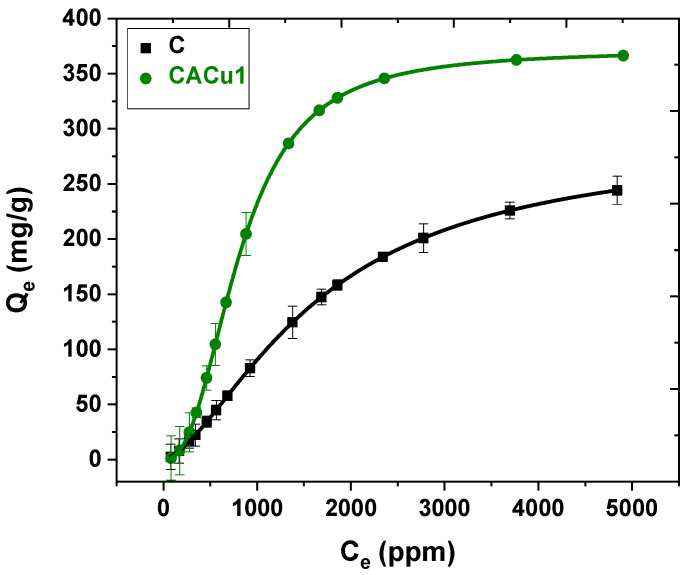
Sips isotherm profiles of C and CACu1 at 298 K and pH 6.8 for adsorption of sulfate from aqueous solution.

**Figure 7 polymers-12-01502-f007:**
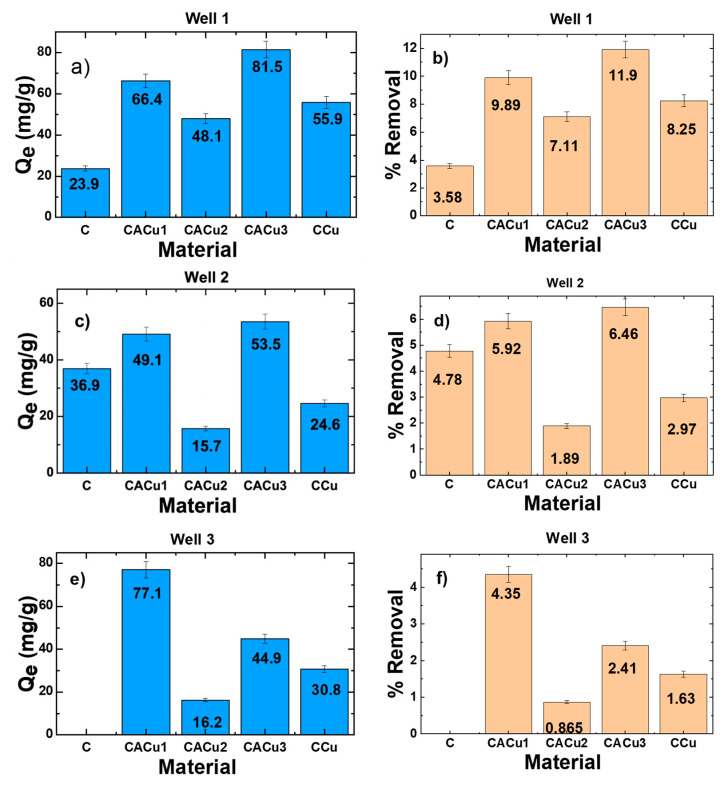
(**a**–**f**) Experimental results of Q_e_ and sulfate removal (%) by five sorbents in Well-1, -2 and -3 groundwater samples.

**Figure 8 polymers-12-01502-f008:**
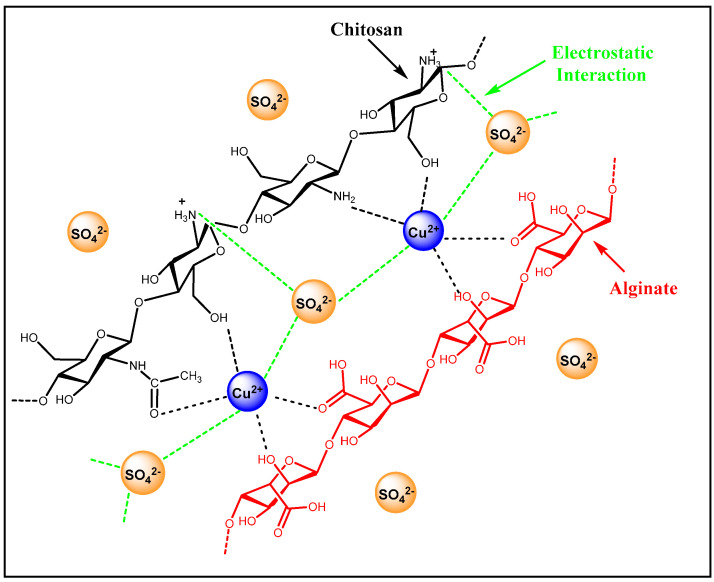
Proposed adsorption mechanism of SO_4_^2−^ anions by a ternary biopolymer composite that contains Cu(II).

**Table 1 polymers-12-01502-t001:** Summary of various chitosan/alginate-based sorbent materials reported in the literature for the removal of SO_4_^2−^ species from aqueous solution.

Adsorbent	Year	References
Chitin-based shrimp shells	2003	Moret and Rubio [53]
Biopolymeric alginate/clinoptilolite pellets	2009	Chmielewská et al. [54]
Protonation modified chitosan	2014	Guo et al. [55]
Chitosan-grafted-polyacrylamide	2015	Fosso-Kankeu et al. [56]
Chitosan-bentonite composite	2016	Mbakop et al. [6]
Chitosan flakes	2018	Schwarz et al. [57]
Magnetic chitosan microspheres	2019	Tian et al. [58]

**Table 2 polymers-12-01502-t002:** Mass of reactants used in the preparation of binary and ternary composite materials.

Reaction Conditions	CCu	CACu1	CACu2	CACu3
Mass of Alginate (g)	0	1.3	2.6	1
Mass of Chitosan (g)	1	1	1	2
Mass of CuCl_2_ (g)	0.83	0.83	0.83	0.83

**Table 3 polymers-12-01502-t003:** TGA parameters for the differential scanning calorimetry (DSC) thermogram profiles for chitosan and its composite materials.

Material	*T*_max_ (°C)	*T*_0_ (°C)	DSC Peak Area (J g^−1^)
C	97.5	59.1	45.9
CCu	135.5	70.3	811.4
CACu1	130.8	52.7	2016
CACu2	125.3	51.9	1491
CACu3	146.6	120.8	786.6

Note: *T*_max_ = maximum peak temperature; *T*_0_ = onset temperature.

**Table 4 polymers-12-01502-t004:** Atom content (At. %) of CACu1 from the XPS wide scan spectra.

Element	Well-1 (At. %)	Well-2 (At. %)	Well-3 (At. %)
C	56.35	53.35	54.01
N	2.14	2.27	1.42
O	27.27	25.99	25.89
Cu	1.17	1.19	1.42
S	-	-	-

**Table 5 polymers-12-01502-t005:** Anion concentration and total hardness (calculated) of the groundwater samples.

Water Composition	Well-1 (mg/L)	Well-2 (mg/L)	Well-3 (mg/L)
Sulfate	2062.6	2653.7	6030.0
Nitrate	6.9	27.6	1.6
Chloride	98.3	146.8	328.1
Bicarbonate	422	593	722
Total Hardness	1912	2198	4194

Note: additional compositional analysis is provided in the Appendix A.

**Table 6 polymers-12-01502-t006:** Comparison of adsorption capacity Q_m_ (mg/g) of various sorbents for the removal of SO_4_^2−^ anions from aqueous solution.

Adsorbent	Q_m_ (mg/g)	Reaction Conditions	References
Pulp and paper waste	2.786	Adsorbent: 40 g Mixture SO_4_^2−^/Cl^−^ 1000 mL pH: not reported Time: 480 h	Lakovleva [72]
Alkali-treated fly ash	43.0	Adsorbent: 2 g SO_4_^2−^ sol^n^: 1000 mL pH: 7 Time: 2.5 h	Geethamani [79]
Organo-nano-clay	63.5	Adsorbent: 500 mg SO_4_^2−^ sol^n^: 100 mL pH: 7 Time: 2 h	Wei and Hai-cheng [80]
Ba-modified blast-furnace-slag geopolymer	119	Adsorbent: 10 g SO_4_^2−^ sol^n^: 1000 mL pH: 7–8 Time: 3 h	Runtti [81]
Chitin-based shrimp shells	250.7	Adsorbent: 750 mg SO_4_^2−^ sol^n^: 200 mL pH: 7 Time: 1 h	Moret and Rubio [44]
Chitosan power (C)	288.1	Adsorbent: 20 mg SO_4_^2−^ sol^n^: 15 mL pH: 6.8–7 Time: 24 h	This study
CACu1	371.4	Adsorbent: 10 mg SO_4_^2−^ sol^n^: 15 mL pH: 6.8 Time: 24 h	This study

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
