# Peer review of "Sequestration of Sulfate Anions from Groundwater by Biopolymer-Metal Composite Materials"

_polymers, 2020, doi:10.3390/polym12071502_

Round 1

Reviewer 1 Report

Dear editor,                                                                                        

The manuscript submitted by Md. Mehadi Hassan et al. entitled: “Chitosan, alginate and copper based composite materials for uptake of sulfate from groundwater environments” aims to investigate composite material using chitosan/alginate for environment field. All things considered it is an interesting study, which can be accepted for publication after major revision. Please to see my comments below.

Comments:

- This title is too confused for reader. Author must reformulate the title of this publication in the revised manuscript.

- In introduction part, authors have made statements and not backed it up with last good references concerning the applications of biopolymers and composite material and more especially using polysaccharides such as chitosan and alginate. Then, authors must give the importance to work on biopolymer for environmental point of view.

- In introduction part, authors should remove the table 1. This table have to be proposed in discussion part of the revised manuscript.

- In result part, the data given were not really explained and discussed for each application. Authors just made a very shallow presentation of their results but not given comparison with literature to give the real impact of the results presented in this work using chitosan/alginate.

- In the conclusion part, the aspects of novelty and the environmental applications of chitosan/alginate composite should be more underlined.

General comment:

  • In the revised manuscript, the authors need to pay more attention to grammatical construction of sentences and spelling of sentences!
  • Line 112, please to change table 2 by table 1.
  • In the revised manuscript, authors must improve the quality of figures and equations.

Author Response

Reviewer Response Document on MS ID polymers-839868

Reviewer #1

Dear editor,                                                                                        

The manuscript submitted by Md. Mehadi Hassan et al. entitled: “Chitosan, alginate and copper based composite materials for uptake of sulfate from groundwater environments” aims to investigate composite material using chitosan/alginate for environment field. All things considered it is an interesting study, which can be accepted for publication after major revision. Please to see my comments below.

Comments:

- This title is too confused for reader. Author must reformulate the title of this publication in the revised manuscript.

Author Response:

To address the reviewer concern, a modified title is proposed: “Sequestration of Sulfate Anions from Groundwater by Biopolymer-Metal Composite Materials”

- In introduction part, authors have made statements and not backed it up with last good references concerning the applications of biopolymers and composite material and more especially using polysaccharides such as chitosan and alginate. Then, authors must give the importance to work on biopolymer for environmental point of view.

Author Response:

The introduction is now revised as recommended. New references have been added to comments on the applications of polysaccharide-based composite materials as recommended.

- In introduction part, authors should remove the table 1. This table have to be proposed in discussion part of the revised manuscript.

Author Response:

We thank the reviewer for this constructive comment. However, we believe that Table 1 provides a context in the introduction helps by highlighting selected previous work and knowledge gaps related to the use of biopolymer composites for the removal of sulfate from water. By contrast, Table 4 shows a selected summary of advanced materials reported from the literature for the adsorption of sulfate. As well, Table 4 highlights the adsorption capacity and provides a comparative view of the sulfate adsorption properties of materials from the present study with selected literature.

- In result part, the data given were not really explained and discussed for each application. Authors just made a very shallow presentation of their results but not given comparison with literature to give the real impact of the results presented in this work using chitosan/alginate.

Author Response:

We thank the reviewer for this comment. The data in Table 5 was to provide an easily accessible way of comparing the results from literature with the results from this author. Furthermore, reference was made to the Table in the revised manuscript.

- In the conclusion part, the aspects of novelty and the environmental applications of chitosan/alginate composite should be more underlined.

Author Response:

We thank the reviewer for this comment. The conclusion was revised to address the reviewer concern, as recommended.

The authors wish to acknowledge Reviewer #1 for the insightful and constructive comments on the above manuscript. We have further edited the manuscript for language, clarity, and syntax throughout to meet the high publication standards of this journal.

Reviewer 2 Report

The paper "Chitosan, alginate and copper based composite materials for uptake of sulfate from groundwater environments" concerns composite materials involving biopolymers and copper(II) inos displaying the sorption capacities for sulfate anions in groundwater.

In general,  I think this article is written in a very good, careful way, it is very rare nowadays and deserves to be treated very well.

Minor comments:

The authors did not define what they maen by "binary and ternary composite materials: that contain animal and/or plant biopolymers (chitosan and alginate) as building blocks in conjunction with copper (II) species. Binary and ternary composite materials should be precisely predined.

Both in the title and in whole text the authors ignore the chemical nature of copper. This must be precisely defined what we are dealing with: ions (oxidation state), metallic structure (nanostructure), etc.

Moreover, the authors make a school mistake by typing "space" between copper and (II). This should be corrected.

In sorption study, I do not understand why the authors compare activity of CACu only with pure chitosan and not with chitosan with Cu(II) material?

In fig 8 the authors should add measurements errors and/or any statistic analysis.

After these minor corrections I recommend paper for publication in Polymers

Author Response

Reviewer Response Document on MS ID polymers-839868

Reviewer #2

The paper "Chitosan, alginate and copper based composite materials for uptake of sulfate from groundwater environments" concerns composite materials involving biopolymers and copper(II) inos displaying the sorption capacities for sulfate anions in groundwater.

In general, I think this article is written in a very good, careful way, it is very rare nowadays and deserves to be treated very well.

Minor comments:

The authors did not define what they maen by "binary and ternary composite materials: that contain animal and/or plant biopolymers (chitosan and alginate) as building blocks in conjunction with copper (II) species. Binary and ternary composite materials should be precisely predined.

Author Response:

We thank the reviewer for this constructive comment. Herein, binary refers to two components (e.g., one biopolymer + Cu(II)), while ternary refers to a 3 component system (e.g. two biopolymers + Cu(II)). That is why the authors included the two and three components that make up the composites in parenthesis after the respective terms and additional comments throughout to avoid any ambiguity or misunderstanding.

Both in the title and in whole text the authors ignore the chemical nature of copper. This must be precisely defined what we are dealing with: ions (oxidation state), metallic structure (nanostructure), etc.

Author Response:

We thank the reviewer for this comment, the authors have now used the phrase “Cu(II) species to denote the fact that copper could be present mostly as divalent ions or oxides in the composites.

Moreover, the authors make a school mistake by typing "space" between copper and (II). This should be corrected.

Author Response: The above noted error has been corrected throughout the revised manuscript.

In sorption study, I do not understand why the authors compare activity of CACu only with pure chitosan and not with chitosan with Cu(II) material?

Author Response:

We thank the reviewer for this comment, the authors only compared chitosan with CACu1 to show the effects of composite formation on the adsorption properties of chitosan. Alginate could not be used as a heterogeneous adsorbent in the same manner as chitosan due to its high solubility in water. Also, CACu1 exhibited the best sorption capacity of sulfate with well water 3 which had the most complex matrix.

In fig 8 the authors should add measurements errors and/or any statistic analysis.

Author Response:

We thank the reviewer for this comment, error bars are now added to the figures in the revised manuscript.

After these minor corrections I recommend paper for publication in Polymers

The authors wish to acknowledge Reviewer #2 for the insightful and constructive comments on the above manuscript. We have further edited the manuscript for language, clarity, and syntax throughout to meet the high publication standards of this journal.

Reviewer 3 Report

Hasan et al. prepared binary and ternary composite materials for sulfate adsorption. The motivation of the work is clear. However, the work has many technical and experimental gaps.

Please find the comments below:

Herein some technical comments:

  1. The manuscript has grammatical errors and spelling mistakes, such as line 14-15 no verb is there.
  2. All the figures need to be carefully drawn with proper caption and marking. The manuscript is really hard to follow.

Experimental gaps:

  1. Cu itself is toxic. Therefore, a leaching study must be provided. 
  2. The stability of the materials must be addressed.
  3. As this is an adsorption based paper, so, all the materials should be used for sulfate removal and finally comparison would be needed for adsorption mechanism.
  4. Raman spectra, 13 C NMR, and XPS need to explain concisely.

Author Response

Reviewer Response Document on MS ID polymers-839868

Reviewer #3

Hassan et al. prepared binary and ternary composite materials for sulfate adsorption. The motivation of the work is clear. However, the work has many technical and experimental gaps.

Please find the comments below:

Herein some technical comments:

1. The manuscript has grammatical errors and spelling mistakes, such as line 14-15 no verb is there.

Author Response:

The manuscript has been revised and all grammatical errors and spelling mistakes are corrected in the revised manuscript.

 2. All the figures need to be carefully drawn with proper caption and marking. The manuscript is really hard to follow.

Author Response:

All figures have been revised as recommended.

Experimental gaps:

3. Cu itself is toxic. Therefore, a leaching study must be provided. 

Author Response:

The authors have not carried out a full characterization of copper concentration was done, using ICP-OES measurements. We demonstrate that insignificant leaching and copper concentrations below 1000 µg/L The related data was added as Table SI-5  to the Supporting Information.

4. The stability of the materials must be addressed.

Author Response:

The stability of the materials was studied through TGA and XPS studies. The result of the studies affirm that the materials only starts degrading above 200 degrees Celsius, while XPS studies revealed that the composition of the materials did not change after sorption studies. The negligible leaching of Cu(II) species corroborates the conclusion related to the XPS results.

5. As this is an adsorption based paper, so, all the materials should be used for sulfate removal and finally comparison would be needed for adsorption mechanism.

Author Response:

All of the materials were used for the sorption of sulfate from the three groundwater samples (well-1, well-2 and well-3). CACu1 displayed the highest removal of sulfate from well-3. It should be noted that well-3 had the most complex matrix and the highest concentration of sulfate. This informed the use of CACu1 for isotherm studies and the proposed adsorption mechanism based on CACu1. These trends have been elaborated upon in the revised manuscript.

6. Raman spectra, 13 C NMR, and XPS need to explain concisely.

Author Response:

The spectral results for Raman, 13 C NMR and XPS were revised as recommended in the revised manuscript.

The authors wish to acknowledge Reviewer #3 for the insightful and constructive comments on the above manuscript. We have further edited the manuscript for language, clarity, and syntax throughout to meet the high publication standards of this journal.

Round 2

Reviewer 1 Report

Dear editor,

Authors improved the revised version.

Consequently, this manuscript could be accepted for publication.

Regards

Reviewer 3 Report

Most of the comments were considered by the authors and the manuscript quality is increased. However, it would be meaningful, if authors use proper labeling in the figures.